# Neurological Erdheim–Chester Disease Manifesting with Subacute or Progressive Cerebellar Ataxia: Novel Case Series and Review of the Literature

**DOI:** 10.3390/brainsci13010026

**Published:** 2022-12-22

**Authors:** Vittorio Riso, Tommaso Filippo Nicoletti, Salvatore Rossi, Maria Gabriella Vita, Perna Alessia, Daniele Di Natale, Gabriella Silvestri

**Affiliations:** 1Department of Neuroscience, Neurology Section, Università Cattolica del Sacro Cuore, 20123 Rome, Italy; 2Dipartimento di Neuroscienze, UOC Neurologia, Ospedale Belcolle, 01100 Viterbo, Italy; 3Department of Neurology, University Hospital Zurich, 8091 Zurich, Switzerland; 4UOC Neurologia, Fondazione Policlinico Universitario A. Gemelli IRCCS, 20123 Rome, Italy

**Keywords:** Erdheim–Chester disease, histiocytosis, neurohistiocytosis, cerebellar ataxia

## Abstract

Neurological involvement is relatively common in Erdheim–Chester disease (ECD), a rare clonal disorder of histiocytic myeloid precursors characterized by multisystem involvement. In ECD patients, neurological symptoms can occur either at onset or during the disease course and may lead to various degrees of neurological disability or affect patients’ life expectancy. The clinical neurological presentation of ECD often consists of cerebellar symptoms, showing either a subacute or progressive course. In this latter case, patients manifest with a slowly progressive cerebellar ataxia, variably associated with other non-specific neurological signs, infratentorial leukoencephalopathy, and cerebellar atrophy, possibly mimicking either adult-onset degenerative or immune-mediated ataxia. In such cases, diagnosis of ECD may be particularly challenging, yet some peculiar features are helpful to address it. Here, we retrospectively describe four novel ECD patients, all manifesting cerebellar symptoms at onset. In two cases, slow disease progression and associated brain MRI features simulated a degenerative cerebellar ataxia. Three patients received a definite diagnosis of histiocytosis, whereas one case lacked histology confirmation, although clinical diagnostic features were strongly suggestive. Our findings regarding existing literature data focused on neurological ECD will be also discussed to highlight those diagnostic clues helpful to address diagnosis.

## 1. Introduction

The term histiocytosis refers to rare, clonal neoplasms derived from macrophage/dendritic cell lineages, giving rise to mutated monocytes [1]: according to the currently proposed pathogenic model, these cells are released in the bloodstream, and then reach the peripheral tissues, where they differentiate into foamy histiocytes eventually causing organ damage, either directly by colonization or indirectly by triggering a pro-inflammatory cascade [2].

Histiocytoses usually affect various tissues and organs, including the bone, the kidney and related retroperitoneal space, the lung, the skin, and the cardiocirculatory and the central nervous systems [1,2,3]. They can either affect children or adults, with differences in individual tissue involvement and histological subtype [4]. According to the most recent classification, five different main forms of histiocytosis are known: (1) Langerhans-related, (2) cutaneous and mucocutaneous, (3) malignant histiocytosis, (4) Rosai–Dorfman disease, and (5) hemophagocytic lymphohistiocytosis and macrophage activation syndrome [2]. Of note, the first group includes both Langerhans cell histiocytosis (LCH) and Erdheim–Chester disease (ECD), previously recognized as distinct entities; nearly 20% of ECD patients also present LCH lesions, and both forms are often associated with clonal pathogenic variants in genes of the MAPK pathway [2,3,4,5,6,7]. LCH mostly occurs in young patients, and ECD in adults; both affect similar brain areas manifesting with related symptomatology [4].

The clinical spectrum of ECD is heterogeneous, ranging from organ-limited (i.e., asymptomatic bone involvement) to disseminated, life-threatening forms; infiltrative lesions commonly involve the long bones determining areas of osteosclerosis, skin (xanthelasma-like lesions), retroperitoneum (peri-renal fat infiltration), cardiovascular system (peri-aortic infiltration, right atrium pseudotumor), orbits (exophthalmos), lungs, hypothalamic–pituitary involvement (diabetes insipidus), and the brain [8,9,10,11,12]. 

In most retrospective studies, clinical signs or symptoms of CNS involvement have been reported in about 40% of ECD patients [13], and in about 25% of the cases they represent the onset and/or the only clinical manifestation of ECD. Overall, brain MRI lesions are detected in more than two-thirds of patients [11], and their prevalence increases to 92% when considering ECD patients with neurological symptoms [13,14]. The clinical presentation of neurological ECD depends both on the site and the type of lesion, but progressive cerebellar and pyramidal symptoms are the most frequent, as CNS lesions in ECD more often affect the cerebellum and the brainstem. Three distinct brain MRI patterns have been recognized in ECD [15]: (i) an infiltrative pattern, with widespread lesions, nodules, or intracerebral masses, mainly involving both cerebellar and brainstem white matter, often without edema or contrast enhancement, (ii) a meningeal pattern with pseudo-granulomatous lesions involving the dura or meningioma-like lesions, and (iii) a composite pattern. Infiltrative lesions of the skull or the pituitary region, and variable signs of atrophy or iron deposition are also not infrequent [13,14,15,16]. Overall, addressing a diagnosis of neurological ECD may be challenging, particularly in patients with isolated CNS involvement manifesting with degenerative-like ataxia phenotypes: therefore, in order to dissect this topic, we describe the results of a retrospective study on a series of four novel neurological ECD patients diagnosed at our Neurological Center between 2009 and 2018. All of them presented with either subacute or chronic cerebellar symptoms associated with prominent infratentorial lesions classified as of unknown etiology. Results will be also discussed in view of available literature.

## 2. Materials and Methods

### 2.1. Patients

We retrospectively reviewed clinical, radiological, and laboratory data of four adult ECD patients (Pts #1–4, age range 52–66 years) diagnosed at the Neurological Unit of Fondazione Policlinico A. Gemelli IRCCS, Rome (Italy) between January 2009 and November 2018. The study was carried out in compliance with the Declaration of Helsinki and approved by the local ethics committee; all patients gave a written informed consent authorizing the storage and use of clinical data also for research studies.

In all patients, neurological ECD manifested with symptoms of cerebellar and/or brainstem involvement and related structural brain MRI lesions of undefined nature. An extensive diagnostic assessment documented typical extra-neurological features of ECD in all patients, and diagnosis of neurological ECD was confirmed in three patients by histopathology studies, whereas based on available clinical and diagnostic findings, the fourth patient was strongly suspected to have ECD. 

### 2.2. Radiological Assessment

All patients underwent at least one conventional brain MR imaging study with contrast medium (gadolinium) on a 1.5 Tesla. All patients also underwent whole skeletal X-rays of upper and lower limbs. Three patients also received technetium 99-metastable (99mTc) bone scintigraphy and thoraco-abdominal CT scans with and without contrast medium, and two of them also underwent whole-body (18F)-fluorodeoxyglucose (FDG) PET. 

### 2.3. Laboratory Assessment 

All patients underwent blood cell count, electrolytes, liver and kidney function tests, serum and urine osmolality, FSH, LH, testosterone/estradiol, ACTH, cortisol, TSH, fT4, prolactin, IGF-1, C-reactive protein (CRP), erythrocyte sedimentation rate (ESR), and serum electrophoresis. CSF examination included chemical, microbiological, and cytology analyses. All patients with more progressive forms also performed an extensive screening for systemic autoimmune markers (including ANA, ENA screen, anti-dsDNA, rheumatoid factor, p-ANCA, and c-ANCA), antigliadin, and anti-onconeural antibodies. Finally, molecular testing for SCA1, 2, and FXTAS was performed in the two patients with a slowly progressive course (Pts 3 and 4).

### 2.4. Other Neurological Diagnostic Tests 

Cognitive functions were assessed by the Mini-Mental State Examination [17] and by the Mental Deterioration Battery [18]. For three out of four patients, upper and lower limb somatosensory (SSEP) and motor evoked potentials (MEP), electromyography, and nerve conduction studies were performed. 

### 2.5. Histopathology

Three patients (Pts 1–3) underwent diagnostic tissue biopsies from femoral, cerebellar and tibial lesions respectively, under general anesthesia: formalin-fixed paraffin-embedded tissues samples were then processed for diagnostic histopathology, including immunohistochemistry for S100, SMA, CK, AE1/AE3, CD45, CD68, CD1a, CD117, CD207, CD21, CD23, MPO, GFAP, desmin, EMA, and OLIG2, and reviewed by experienced pathologists. Iliac crest bone marrow needle aspiration and biopsy were also performed in Pts 1 and 2. 

## 3. Results

### 3.1. Clinical Features

Demographic, clinical, and diagnostic data of the four ECD patients are summarized in Table 1. All patients were Caucasian; 75% were male. The median age at the onset of symptoms was 60.2 years (range 52–66 years). The mean follow-up duration was 4.62 years (standard deviation 3.51 years, range 1–8 years). 

All patients manifested with either subacute or slowly progressive cerebellar symptoms mainly consisting of gait ataxia and dysarthria; three patients (75%), had also upper limb dysmetria and gaze-evoked nystagmus, and instability for subjective vertigo and ophthalmoparesis were evident in one patient (25%). 

Pyramidal signs were also present in one patient (25%) who showed a mild right brachiocrural hemiparesis, and two patients (50%) had brisk deep tendon reflexes, and showed presented variable signs of involvement of cranial nerves (VII, VIII, and X). 

Extrapyramidal signs were evident in two patients (50%). 

Cognitive impairment in the form of attention/dysexecutive deficit was evident in one patient (25%) at the onset of symptoms. 

Notably, pseudobulbar affect appeared very early in the clinical course of all three patients with more progressive symptoms.

Two patients (Pts 1 and 4) were lost to long-term follow-up, although Pt 1, who was treated by conventional chemotherapy after diagnosis of ECD, was noted to be neurologically stable after two years. 

Patient 2 is currently in follow-up. He is now not able to maintain either standing or sitting positions and recently manifested seizures that were controlled by levetiracetam. His last neurological examination documented severe spastic ataxia, with marked right hemiparesis, axial, and appendicular ataxia, nystagmus, and cerebellar dysarthria, partially due to neurological sequelae after a diagnosis of cerebellar histiocytic sarcoma and related treatment. 

Patient 3 manifested during the follow-up a mild worsening of cerebellar ataxia and cognitive functions at neuropsychological tests. He eventually died of pneumonia six years after the onset of symptoms. 

Patient 4, who had the longest period of follow-up (8 years), had moderate swallowing disturbances eight years after the initial assessment postulating a diagnosis of neurological ECD. 

### 3.2. Laboratory Examinations

Cell blood count was normal in all patients, as well as electrolytes, liver, and renal function tests, except for slight elevated creatinine level in Pt 1, who also had increased blood IgE (506 UI/mL) and low-titer positive anti-ANA antibodies (1:160 dilution). Pituitary function and other immunological tests were normal in all cases. The CSF examination revealed slightly increased protein levels only in Pts 3 and 4, while other chemical parameters, microbiological and immunological analyses were otherwise unremarkable in all patients (Table 1). 

### 3.3. Neuroimaging

Brain MRI showed T2/FLAIR hyperintense signal alterations without contrast enhancement variably involving the white matter of cerebellar hemispheres, middle cerebellar peduncles, dentate nuclei, pons, and midbrain and cerebral peduncles in all cases (Figure 1). Three patients (Pts 1, 2, 4) also presented similar supratentorial T2/FLAIR hyperintensities variably involving the paratrigonal area, posterior arm of the internal capsule, globus pallidus, peri-aqueductal, and parahippocampal areas, with evidence of gadolinium contrast enhancement in the paratrigonal area not shown in Pt 3. Moreover, signs of cerebellar atrophy and bilateral hypointense signal abnormalities in basal ganglia on SWI sequences, suggestive of iron deposition, were evident in Pts 1, 3, and 4 (Figure 1). Of note, in Pt 2 manifesting a subacute neurological outcome, MRI also showed two nodular lesions (30 × 20 × 20 mm^3^) in the right cerebellar hemisphere with contrast enhancement (Figure 2) and elevated levels of r-CBV (regional cerebral blood volume) on the perfusion study and DWI-restriction, indicative of hypercellularity. Finally, Pt 3, affected by diabetes insipidus, also presented an empty sella. Signs of paranasal sinuses or mastoid inflammation were evident in three patients (Pts 1, 2, and 3). Spinal cord MRI was normal in all patients.

### 3.4. Neurophysiology 

SEP and MEP documented damage of both central sensory and motor pathways in all patients. Electromyography and nerve conduction studies were normal in two out of three patients assessed, whereas in the other patient (Pt 2), also affected by diabetes mellitus type 2, documented a mild predominantly demyelinating lower limb polyneuropathy. 

### 3.5. Cognitive Studies 

Regarding cognition, initial diagnostic evaluation in Pt 1 manifested impairment in attention, executive, memory, and language (verbal fluency); similarly, Pt 4 presented a mild impairment in the same cognitive domains. In Pts 3 and 4, cognitive test were initially normal, and a serial evaluation three years later documented only in Pt 3 a moderate worsening in attention, and executive, memory, and language functions (verbal fluency). Finally, Pt 2 showed a mild defect only in one test assessing verbal fluency at the initial cognitive assessment.

### 3.6. Systemic Involvement

In the differential diagnosis for ECD, all patients underwent whole-body skeletal X-ray, which showed multiple areas of bone osteosclerosis, mainly involving the diaphyseal sections of upper and lower limb long bones, ribs, and the thoracic vertebrae in three patients (Pts 1, 2, and 3) (Table 1).

These patients underwent further nuclear imaging studies (either 18-FDG CT-PET or Tc99m bone scintigraphy), all of which documented pathological uptake of the radiotracers in correspondence of the osteosclerotic lesions. Notably, increased tracer uptake was detectable in other bone sites (jaw, hip bone, vertebral column, heels) that were apparently unaffected at X-ray examination.

As previously pointed out, one patient (Pt 3) had a history of diabetes insipidus and was receiving treatment with nasal desmopressin: laboratory findings showed normal values of plasmatic and urinary osmolarity, with otherwise conserved function of the hypothalamus–hypophysis axis. Another patient (Pt 2) was affected by diabetes mellitus type 2 and treated with oral glucose-lowering agents.

A chest CT scan showed imaging features of bronchiolitis in Pt 3, while Pts 1 and 2 had a “ground glass” appearance in the lungs, also associated with slight bilateral pleural effusion and apical pleural thickening in Pt 1. Two patients (Pts 1 and 3) showed retroperitoneal fibrosis with “hairy kidney” aspect at the abdominal CT scan, although with conserved renal function. Another patient (Pt 2) had small cortical cysts bilaterally with edema of perinephric bridging septa (Table 1). An abdominal echoscan was also normal in Pt 4. Aortic parietal calcium deposits were documented in all patients. 

Echocardiography was normal in all patients. Finally, eyelid xanthelasmas were evident in Pt 4.

### 3.7. Histopathology Data

Biopsy sites were the cerebellum in Pt 2 and the long bones in Pts 1 and 3 (tibia and femur, respectively). In all samples, numerous histiocytes were found, either multinucleated or with foamy or vacuolated cytoplasm identifiable by their immunohistochemical expression of CD68 and absence of S-100 and CD1a, which was associated with the presence of abundant fibrotic tissue or lymphoplasmacellular infiltration with positive anti-CD45 immunoreactivity. Histiocytes from Pt 1 also showed CD117 and CD207 immunoreactivity, which was instead absent in the cerebellar sample from Pt 2: in this case, mitosis and an increased proliferative index (MIB 1 = 10–15%) were also evident. According to its histopathology features, the cerebellar nodule was diagnosed as a histiocytic sarcoma. 

Histochemistry for BRAFV600E was negative in the bone biopsy of Pt 1. Results of molecular testing for the BRAFV600E mutation were not available from medical records in Pt 2, while testing was negative in whole leukocytes DNA from Pt 3.

### 3.8. Management and Follow-Up

Following diagnosis, both Pts 1 and 2 were referred to the Division of Haematology of our Centre for Therapeutic Management and Follow-up. Pt 1 was initially treated with oral steroids without any clinical improvement and then treated with chemotherapy by subcutaneous cladribine (5 injections/month for 4–5 months). Serial PET FDG studies showed persistent reduced uptake of the radioligand in correspondence of the bone lesions during two years of follow-up. 

Pt 2, diagnosed with histiocytic sarcoma, was treated with radiotherapy and chemotherapy (temozolomide 150 mg for 3 months) with poor response; thus, he underwent an autologous peripheral stem cell transplantation, preceded by a FEAM (fotemustine/aracytine/etoposide/melphalan) protocol as conditioning therapy. Serial MRI and scans showed good response to treatment, with no evidence of disease recurrence at both MRI and global PET CT scan two years after treatment.

According to their clinical histories and the results of the diagnostic work-up, both Pts 3 and 4 were diagnosed with neurodegenerative ECD. Based on the evidence of contrast enhanced lesions in the brain MRI, Pt 3 underwent a cycle of steroid pulse therapy (1 g/day for 5 days) followed by a maintenance therapy with oral corticosteroids (starting dose of prednisone 50 mg/day), which produced a temporary, partial clinical benefit on gait balance problems, with a concomitant disappearance of contrast enhancement in the brain MRI (data not shown). However, his cerebellar ataxia thereafter returned to slowly progressive despite this treatment, so steroids were tapered until suspension, without registering any significant clinical and neuroimaging worsening. Patient 4 had a very mild disease progression. In consideration of the relatively benign course, and of the potential side effects of conventional therapies, no further treatment was performed for either patient. 

## 4. Discussion

This case series illustrates the occurrence of Erdheim–Chester disease presenting exclusively with neurological manifestations. This issue now appears important, as CNS involvement may affect the life expectancy of ECD patients [19], and recent research indicates there is a significant response of CNS symptoms to novel drugs specifically targeting the MAPK pathway [14,20], the dysregulation of which plays a main role in the pathogenesis of ECD. The relevance of CNS involvement in ECD is underlined by different systematic literature reviews, regarding both the high prevalence of neurological involvement and also of neurological presentation of ECD. 

Of note, in our small ECD cohort neurological involvement remained the only clinical manifestation of ECD, consisting of subacute or progressive cerebellar and brainstem symptoms associated with prominent involvement of the infratentorial white matter on the brain MRI. A similar presentation was previously highlighted in a previous case series [21], and our findings actually support that patients with exclusive neurological ECD might represent a specific subgroup, distinct from ECD, characterized by extraneurological presentation. 

According to the literature, we observed a similar male prevalence in our cohort (3M, 1F). In addition to cerebellar symptoms, the pseudobulbar effect was another common clinical feature in our ECD cohort; accordingly, in a recent review of 30 ECD patients with prominent neurological involvement, Bathia and colleagues [14] reported the presence of bulbar affect in about 30% of cases, supporting that this symptom might represent a “red flag” to suspect neurological ECD when combined with ataxia and the presence of peculiar infratentorial neuroimaging lesions. 

Other neurological manifestations, such as cranial nerves involvement or diplopia, similarly occurred at a relatively low prevalence in our small ECD cohort. In addition, 50% of our patients showed clinical pyramidal signs, and in all of them, motor and sensory evoked potential documented an involvement of both long descending and ascending central pathways, likely related to the widespread white matter brainstem involvement. 

A high prevalence of cognitive impairment was also documented in our cohort of neurological ECD patients, mainly affecting verbal fluency, memory, and executive functions. These findings are in agreement with those of Boyd et al. [16], who reported a similar prevalence and pattern of cognitive involvement in a cohort of 15 ECD patients assessed by a detailed neuropsychological battery. In this study, voxel-based morphometry documented significantly reduced brain volumes in ECD patients vs. healthy controls, with specific loss of grey matter in the right frontal and parietal cortex, and routine brain MRI from a larger cohort of 62 ECD patients documented signs of brain atrophy in about 20–30% of cases. These results suggest the occurrence of neurodegenerative damage in ECD brains. Accordingly, brain MRI showed both cerebellar atrophy and signs of iron deposition both in the basal ganglia and cerebellar nuclei in Pts 1, 3, and 4 with a progressive neurological disease course. 

Regarding brain MRI, all our neurological ECD patients had a suggestive “infiltrative pattern”, characterized by T2/FLAIR hyperintense white matter lesions mainly involving the brainstem and the cerebellum, usually without contrast enhancement. Some of them showed similar alterations also in the supratentorial compartment, in one case with focal areas of contrast enhancement. Overall, our data support those of previous reports indicating that the characteristic brain MRI findings in “pure “neurological ECD patients consists of a diffuse, exclusive, or prominent involvement of the infratentorial compartment mainly involving the cerebellum, the brainstem, and the cerebellar peduncles, usually without contrast enhancement [13,14,15,16,21] and with no evidence of meningeal involvement. As an additional diagnostic hint for ECD, diabetes insipidus had occurred in Pt 3 many years before the onset of ataxia. Such a condition is relatively frequent in ECD, being reported in 30–48% of patients, and it may occur, as in our patient, many years before the onset of neurological symptoms. A long-lasting history of diabetes insipidus associated with thickening and contrast enhancement of the pituitary stalk, in the presence of unspecific brain, meningeal, or retro-orbital lesions might also imply a differential diagnosis of IgG4-related disease, sarcoidosis, tuberculosis, or lymphocytic infundibulo-neurohypophysitis. Yet, the association with diffuse signs of long bone involvement is highly suggestive for a diagnosis of neuro-ECD, as discussed shortly after in this section. Finally, two large tumoral nodular lesions in the right cerebellar hemisphere with contrast enhancement were also present in Pt 2, who accordingly manifested rapidly evolving symptoms due to its mass effect in the posterior fossa. 

Another diagnostic clue for neurological ECD might be represented by the presence of signs of sinusitis and/or mastoiditis on the brain MRI, as subclinical systemic disease manifestations, reported in about 30% of ECD patients characterized by neurohystiocitic involvement [14,21]. In this regard, however, the most suggestive extraneurological feature of ECD present in three of our patients was the presence of multiple areas of bone osteosclerosis, mainly involving the diaphyseal sections of long bones, ribs, and the thoracic vertebrae, and characterized by more widespread pathological uptake of the radiotracers at nuclear imaging studies. Asymptomatic bone lesions are reported with a very high prevalence (80–100%) in neuro-ECD cohorts [14,21,22]. However, their absence does not rule out an ECD diagnosis, as about 20–26% of ECD patients may not present signs of bone involvement [8,11]. 

Other typical subclinical systemic ECD signs were detected in Pts 1, 2, and 3 through thoraco-abdominal CT scans: these included signs of renal and retroperitoneal infiltration characterized by hairy kidney and ground glass appearance of lungs or pleural effusion. Moreover, eyelid xanthelasma-like lesions were detected in Pt 4. Conversely, none of our patients presented signs of periaortitis, or heart involvement.

A definite diagnosis of ECD, according to current diagnostic criteria for ECD [20] requiring histology confirmation, was reached in three patients of our cohort: in particular in Pt 2, one nodular lesion configured a histiocytic sarcoma. Histiocytic sarcoma, a malignant proliferation of cells showing signs of histiocytic differentiation, often occurring in the skin, the lymph nodes, and the intestinal tract, while primary peripheral or CNS involvement is very rare [23,24]. It has often poor prognosis, but patients with localized histiocytic sarcoma may survive years after the initial diagnosis followed by aggressive clinical management. Data from single case reports have suggested efficacy for high-dose chemotherapy and autologous/allogeneic stem cell transplantation; indeed, our patient, treated by autologous peripheral stem-cell transplantation preceded by FEAM (fotemustine/aracytine/etoposide/melphalan) protocol as conditioning therapy, shows no signs of active disease at the last CT-PET after 4 years of follow-up. In Pt 2, the presence of systemic bone manifestations, and the evidence of diffuse infratentorial brainstem and cerebellar alterations with only limited contrast enhancement at the brain MRI are both features suggestive of ECD. Thus, we suggest that both ECD and histiocytic sarcoma may coexist in this patient, originating from a common neoplastic precursor, as such disorders may share the same oncogene background [2,6,7,25,26].

In both Pts 1 and 3, bone histology confirmed ECD diagnosis. Following hematological evaluation, Pt 1 was treated by conventional chemotherapy, and we could observe a substantial neurological stability after two years of follow up; then, the patient was lost. After steroids, in agreement with hematologists, Pt 3 was not treated with further drugs, in consideration of his isolated and slowly progressive neurological involvement and the potential serious side effects of other immunosuppressive or chemotherapy drugs. Unfortunately, we did not reach a definite histological confirmation of ECD in Pt 4, yet we believe that her clinical and brain MRI features, together with the presence of skin manifestations, would strongly support an ECD diagnosis also in this case. 

ECD has been associated with mutations in genes involved either in the MAPK and the P13K-AKT signaling pathways, which eventually activate cell proliferation, survival, and angiogenesis; in particular, more than 50% of the ECD patients carry somatic BRAF (V600E) gene mutation in affected tissues, that sometimes can be detected also on DNA from peripheral blood. In our retrospective data analysis, histochemical staining for BRAFV600E was reported to be negative in bone tissue biopsy of Pt 1, and molecular testing, performed only in blood DNA, was also negative in Pt 3. No molecular data regarding Pt 1 and Pt 4 were available from medical records. 

Currently, molecular characterization of ECD lesions may allow the establishment of more effective chemotherapies specifically targeting the dysfunctional pathway, i.e., BRAF inhibitors vemurafenib or dabrafenib for BRAF-mutated ECD patients, and recent studies indicate that these drugs would be also successful on CNS involvement. 

However, the pathogenesis of neurological damage in ECD still needs clarification: indeed, signs of cortical brain atrophy have also been documented in ECD patients without corresponding supratentorial lesions [16,27]. Further studies are needed to better clarify the pathogenesis of neurodegeneration in ECD patients, which might be either triggered by systemic inflammation secondary to histiocytic infiltration in other tissues, or by other still unknown mechanisms [2].

## Figures and Tables

**Figure 1 brainsci-13-00026-f001:**
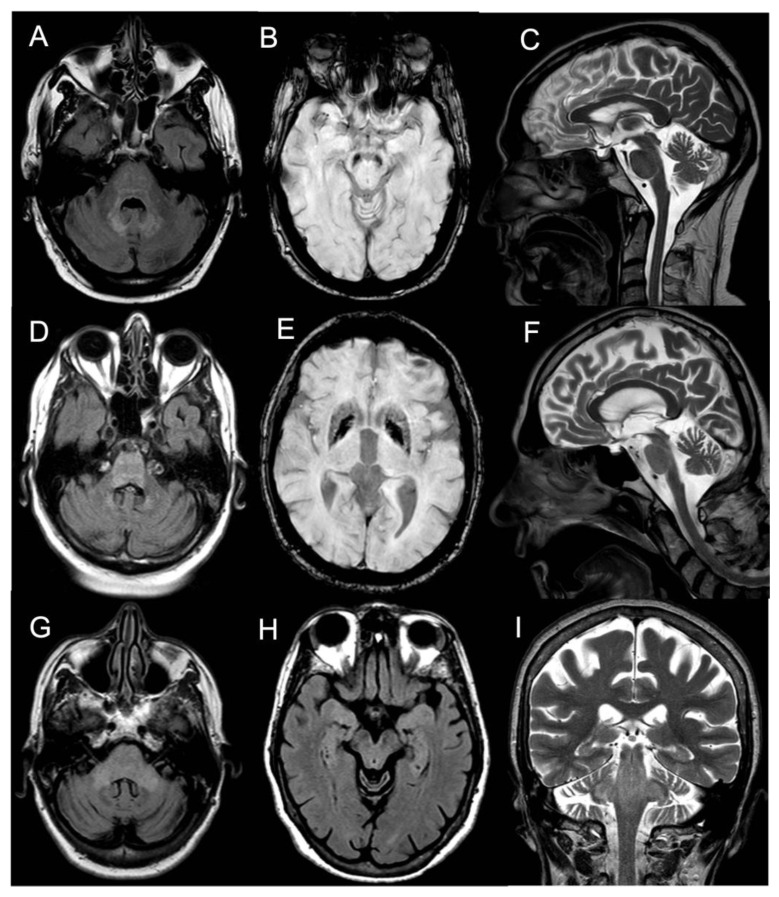
Panel showing brain MRI of Pt #1 (**A**–**C**), Pt #3 (**D**–**F**), and Pt #4 (**G**–**I**). (**A**) Transverse FLAIR scan (pons level) of Pt #1 showing mild hyperintensity of the pons, both MCPs, dentate nuclei. (**B**) Transverse SWAN scan (mesencephalon level) of Pt #1 showing marked hypointense signal (metal deposits) of bilateral substantia nigra. (**C**) Sagittal T2-weighted scan of Pt #1 showing slight vermian cerebellar atrophy. (**D**) Transverse FLAIR scan (pons level) of Pt #3 showing hyperintensity of the pons and dentate nuclei. (**E**) Transverse SWAN scan (thalami level) of Pt #3 showing marked hypointense signal (metal deposits) of putamina and globi pallidi. (**F**) Sagittal T2-weighted scan of Pt #3 showing vermian cerebellar atrophy. (**G**) Transverse FLAIR scan (pons level) of Pt #4 showing hyperintensity of the pons and both MCPs. (**H**) Transverse FLAIR scan (mesencephalon level) of Pt #4 showing hyperintensity of the mesencephalon and parahippocampal cortices. (**I**) Coronal T2-weighted scan of Pt #4 showing hyperintensity of the pons, both SCPs and MCPs. Abbreviations: Flair: fluid-attenuated inversion recovery; MCP: middle cerebellar peduncle; SWAN: susceptibility-weighted angiography; SCP: superior cerebellar peduncle.

**Figure 2 brainsci-13-00026-f002:**
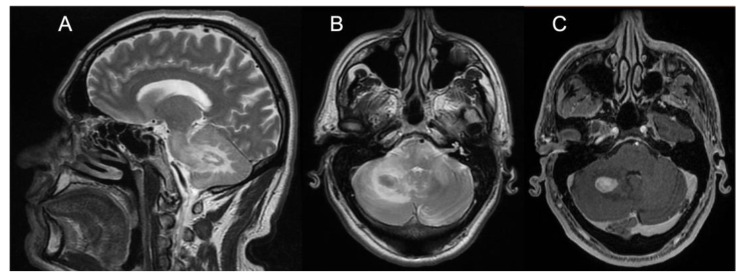
Panel showing brain MRI of Patient #2. (**A**) Sagittal T2-weighted scan showing one nodular lesion along with extensive hyperintensity of the white matter of the RCH, the ipsilateral MCP, and the pons. (**B**) Transverse T2-weighted scan (pons level) showing the same nodular hypointense lesion with hyperintensity of the white matter of the RCH, both MCPs and the pons. (**C**) Transverse T1-weighted scan (pons level) showing marked, homogeneous contrast enhancement of the RCH nodular lesion. Abbreviations: RCH: right cerebellar hemisphere; MCP: middle cerebellar peduncle.

**Table 1 brainsci-13-00026-t001:** Summary of the demographic, main clinical, and diagnostic features documented in our case series and related treatment strategies.

Pt#, Sex	AAO	AE	Neurological	Other Neurological Findings	Skeletal Involvement	Brain MRI	Hystopathology	Treatment	Follow-Up
1,M	52	52	Dysarthria, gait ataxia, subjective dizziness	Parkinsonism, right lateropulsion in Romberg test, gaze-evoked horizontal nystagmus, eyelids ptosis, pseudobulbar affect	Area of osteosclerosis in diaphyseal bones with diffuse 18-FDG uptake	*Infiltrative pattern:* T2/FLAIR hyperintensity in MCP, pons, DN and GP without CE. Iron deposits in GP, DN and SN bilaterally.	*Left tibia biopsy*: Fibrosis with numerous macrophages, some of them foamy, and lymphocytes CD68+/CD45+, S100-, CD1a-, CD117-, CD207-. Immunochemistry for BRAFV600E-.	Subcutaneous Cladribine.	2 years, stable. Then lost at the follow-up.
2,M	60	60	Dysarthria, gait ataxia	Right lateropulsion in Romberg position, gaze- evoked rotational nystagmus, bilateral dysmetria, right hemiparesis	Diffuse osteosclerosis (vertebral body D10, left fifth rib and femur heads) with diffuse 18- FDG uptake	*Composite pattern:* T2/FLAIR hyperintensity in cerebellar hemispheres, MCP, left SCP, pons, bulb, posterior arm of IC. Two nodular areas in right cerebellar hemisphere with hypointensity in long-TR scan and with CE. Small areas of CE in the pons and in the right MCP	*Cerebellar biopsy:* Numerous histiocytes often multinucleated and with cytoplasmic vacuolation. CD 68+, S100-, CD 207-, CD1a-. Fibrosis with CD45+ lymphocytes. Conclusions: histiocytic sarcoma.	Temozolomide, autologous stem cells transplantation (FEAM protocol)	4 years, stable. Last PET-TC negative.
3,M	63	64	Dysarthria, gait imbalance, cognitive impairment	Trunkal ataxia, ophthalmoparesis, gaze- evoked nystagmus, brisk deep tendon reflexes, bilateral dysmetria, upper limbs dystonia, pseudobulbar affect.	Areas of osteosclerosis in diaphyseal bones of the 4 limbs, and II rib. Frontal sinus osteoma. Bone uptake in Tc99m scintigraphy.	*Infiltrative pattern:* T2/FLAIR hyperintensity in pons and midbrain with CE. T1 hyperintensity of pallidus nuclei. Cerebellar atrophy and iron deposits in striatal nuclei. Empty sella.	*Femur biopsy:* numerous macrophages/histiocytes CD68+ and PGM1+, S100-, SMA-.	High-dose steroid pulse therapy followed by oral maintenance therapy	6 years, worsening of ataxia and cognition, dysphagia. Death for respiratory complications.
4,F	66	76	Dysarthria, dysphonia, progressive right facial nerve peripheral palsy. Gait ataxia later	Slowing of saccadic movements, brisk deep tendon reflexes, bilateral dysmetria, pseudobulbar affect.	None	*Infiltrative pattern:* T2/FLAIR hyperintensity in pons, midbrain, cerebellum, periaqueductal and parahippocampal areas without CE. Iron deposits in GP.	Not done.	None	10 years, then lost. During follow-up: dysphagia.

Abbreviations: AAO = age at onset; AE = age at the first neurological examination; MCP = middle cerebellar peduncles; SCP = superior cerebellar peduncle; DN = dentate nuclei; GP = globus pallidus; SN = substantia nigra; CE = contrast enhancement; IC = internal capsule.

## Data Availability

Anonymized patients’ data are available on reasonable request.

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
