# Peer review of "Neurological Erdheim–Chester Disease Manifesting with Subacute or Progressive Cerebellar Ataxia: Novel Case Series and Review of the Literature"

_brainsci, 2022, doi:10.3390/brainsci13010026_

Round 1
Reviewer 1 Report
Review: Neurological Erdheim-Chester disease manifesting with subacute or progressive cerebellar ataxia: novel case series and review of the literature
Occasionally, it is a bit hard to follow, mainly because of the lack of structure in certain paragraphs (in particular for the description of the clinical features.
Please find below only minor comments/suggestions.
ABSTRACT:
No comments.
INTRODUCTION:
Please remove the typo in l. 44 „Langherans“.
METHODS:
Was only single gene sequencing of the three listed genes performed? The list is - by far- not exhaustive for slowly progressive ataxias.
RESULTS:
The clinical features are somewhat described in an odd order. It might make sense to rephrase the respective paragraph.
The table likely needs reformatting. At least in the review version of the manuscript, it is impossible to see the whole „Neurological“ from patient 4.
Please change „SSEP“ to „SEP“ (l. 180).
Please separate the paragraphs „neurophysiological testing“ and „cognitive studies“.
Please use „Diabetes mellitus type 1/2“ throughout the manuscript.
Are there any histopathological images available? IMHO, the diagnosis mainly relies on this test, so it would be nice for the reader to have a figure with the confirmatory results.
DISCUSSION:
No comments.
Author Response
We thank the reviewer for his/her constructive comments and suggestions, that we accepted revising our manuscript accordingly. Please find below point to point answers
Review: Neurological Erdheim-Chester disease manifesting with subacute or progressive cerebellar ataxia: novel case series and review of the literature
Occasionally, it is a bit hard to follow, mainly because of the lack of structure in certain paragraphs (in particular for the description of the clinical features.
Please find below only minor comments/suggestions.
ABSTRACT:
No comments.
INTRODUCTION:
Please remove the typo in l. 44 „Langherans“.
We’ve changed it.
METHODS:
Was only single gene sequencing of the three listed genes performed? The list is - by far- not exhaustive for slowly progressive ataxias.
Only the three listed ADCA genes have been tested, the most frequent in Italy being SCA1 and 2 , and FXTAS because of age at onset and some common radiological features regarding MCP alterations. Further investigations were not performed after diagnostic workup pointed to ECD diagnosis.
RESULTS:
The clinical features are somewhat described in an odd order. It might make sense to rephrase the respective paragraph.
We rephrased the paragraph.The table likely needs reformatting. At least in the review version of the manuscript, it is impossible to see the whole „Neurological“ from patient 4.
We did it.Please change „SSEP“ to „SEP“ (l. 180).
We did it.Please separate the paragraphs „neurophysiological testing“ and „cognitive studies“.
We did it.Please use „Diabetes mellitus type 1/2“ throughout the manuscript.
We did it.
Are there any histopathological images available? IMHO, the diagnosis mainly relies on this test, so it would be nice for the reader to have a figure with the confirmatory results.
Unfortunately, we only receive written response and not pictures of the histopathology studies. We do not have direct access to histology specimens, and so far we did not get any response form the pathologist that we contacted to get some images, taking into account the short time span for review from Brain Sciences ( 5 working days) so we could not provide images
DISCUSSION:
No comments.
Reviewer 2 Report
This manuscript is a case series of 4 patients with apparent Erdheim-Chester disease (ECD) who had a clinical presentation of cerebellar ataxia without systemic manifestations. It is not clear to me how unique this series is, but it does raise some interesting points. There are, however, a significant number of corrections and clarifications that need to be addressed. I will list them as they appear in the manuscript.
Lines 20 and 23: It is not entirely valid to say that these presentations simulated degenerative ataxias, particularly the MRI findings. It would be highly unusual for the types of MRI findings described in these cases to be seen in typical degenerative ataxias, certainly hereditary ones. The most likely confusing entities would be autoimmune ataxias in which discrete inflammatory elements may be recognized by neuroimaging.
Line 37: change to “histiocytoses usually affect”
Line 59: change “raises up” to “increases”
Line 86: Patient #2 was diagnosed pathologically with “histiocytic sarcoma”. Is this considered to be a form of ECD, and if so by whose criteria; or is this considered to be a misdiagnosis? I do not think that there is general agreement on this question in the literature. This issue comes up later in the paper as well.
Line 92: “performed” should be changed to “received” or “underwent”
Line 93: change “scan” to “scans”
Lines 135-136: This sentence is unclear. Change to “Two patients (Pts1 and 4) had been lost to long-term follow-up, although Pt 1, who was treated by conventional chemotherapy after diagnosis of ECD, was noted to be neurologically stable after two years.”
Table on Page 4: The justification is not correct within the cells and several cells appear to have some of the text cut off at the bottom. There also are some discrepancies between what is stated in the table and the text, which I will point out in detail below.
Line 160: I take it that this includes pituitary function in the patient with “empty sella”.
Lines 166-169: Where was the supratentorial enhancement localized in the two patients that had this finding? Either describe it or illustrate it.
Lines 170-171: The SWI sequences show iron deposition in areas known to have high iron content. This is not a truly quantitative test, so how do you know it is abnormal?
Lines 175-176: What is the significance of the “empty sella”? There was DI, which could be related to ECD, but no disfunction of the anterior pituitary is mentioned. Therefore, it is likely that the “empty sella” is not directly related to ECD, i.e., empty sella, without empty sella syndrome.
Lines 197-198: According to the Table on page 4, the lesions in patient 2 did not have uptake of radioactive tracers. This is important because the tissue diagnosis from the cerebellar biopsy was not called “ECD”.
Line 199: change “districts” to “sites”
Lines 201-203: This sentence is confusing. If the patient had DI and required desmopressine, how could the hypothalamic-pituitary-axis function be “conserved”?
Line 207: change “to” to “with”
Line 209: change “at abdomen” to “on abdominal”
Line 210: change “cist” to “cyst”
Line 221: The table on page 4 states that the in Pt 1 the CD207 was negative and does not mention CD117. If CD207 was indeed positive, how often would that occur in ECD and in particular in the face of negative staining for CD1a and S100?
Line 224: See lines 37 and 86 above
Line 233: change “captation” to “uptake” and change “to” to “with”
Line 235: change “radio” to “radiotherapy” (is there any information on the dose delivered?)
Line 241: change “history” to “histories”
Line 242: start at new sentence at the colon
Line 243: change “at” to “in”
Line 246: change “at” to “in”
Line 251: change “both” to “either”
Line 276-281: Awkward wording. Change sentence to “ This case series illustrates the occurrence of Erdheim-Chester disease presenting exclusively with neurological manifestations.” Then start new sentence with “This issue now appears important, as CNS involvement may affect the life expectancy of ECD patients [19], and recent research indicates there is a significant response of CNS symptoms to novel drugs specifically targeting the MAPK pathway [14,20], the dysregulation of which plays a main role in the pathogenesis of ECD.”
Line 286: change “at” to “on”
Lines 291-292: What is the reported age range of onset? Is a cohort of 4 patients adequate to say anything meaningful about a discrepancy?
Line 299: none of your patients had epilepsy, which is a very low prevalence This sentence should be worded better.
Lines 313-315: See Lines 170-171
Line 316: Why is the MRI appearance “peculiar”. I thought it was a subtype of what is expected in ECD (Lines 64-66).
Lines 319-320: Which “case” is being referred to? There were two cases with contrast-enhancement described above. (Lines 166-169) Change “remind” to “reinforce” or “support”
Line 322: change “consists in” to “consists of”
Line 325-326: The empty sella is likely to be a red herring here. The presence of DI is the important finding and does not necessarily follow from the empty sella. You would not get an empty sella from damage to the posterior lobe alone.
Line 343: the results for pt 2 need to be reconciled with the Table on page 4 as noted above
Lines 355-64: the relationship between “histiocytic sarcoma” and ECD needs to be clarified with some support from the literature. It still is uncertain to me whether Pt 2 has ECD or something different. If that patient is to be included, there needs to be more justification as to why?
Line 379: I think this should be patient 1 and not patient 2.
Lines 386-391: There need to be citations supporting some of these statements. What is the evidence for neurodegeneration in the absence of infiltration of histiocytes?
Author Response
We thank the reviewer for his/her exhaustive and constructive comments and suggestions, that we accepted revising our manuscript accordingly. Please find below point to point answers
This manuscript is a case series of 4 patients with apparent Erdheim-Chester disease (ECD) who had a clinical presentation of cerebellar ataxia without systemic manifestations. It is not clear to me how unique this series is, but it does raise some interesting points. There are, however, a significant number of corrections and clarifications that need to be addressed. I will list them as they appear in the manuscript.
Lines 20 and 23: It is not entirely valid to say that these presentations simulated degenerative ataxias, particularly the MRI findings. It would be highly unusual for the types of MRI findings described in these cases to be seen in typical degenerative ataxias, certainly hereditary ones. The most likely confusing entities would be autoimmune ataxias in which discrete inflammatory elements may be recognized by neuroimaging.
We agree with this observation and modified the text following Your suggestion.
Line 37: change to “histiocytoses usually affect”
We changed it.
Line 59: change “raises up” to “increases”
We changed it.
Line 86: Patient #2 was diagnosed pathologically with “histiocytic sarcoma”. Is this considered to be a form of ECD, and if so by whose criteria; or is this considered to be a misdiagnosis? I do not think that there is general agreement on this question in the literature. This issue comes up later in the paper as well.
We explained better this issue and added references to support this hypothesis. In pt #2, the presence of systemic bone manifestations and the evidence of diffuse infratentorial brainstem and cerebellar alterations with only limited contrast enhancement at the brain MRI are both features suggestive of ECD. Thus we suggest that both ECD and histiocytic sarcoma may coexist in this patient originating from a common neoplastic precursor, as such disorders may share the same oncogene background
Line 92: “performed” should be changed to “received” or “underwent”
We changed it.
Line 93: change “scan” to “scans”
We changed it.
Lines 135-136: This sentence is unclear. Change to “Two patients (Pts1 and 4) had been lost to long-term follow-up, although Pt 1, who was treated by conventional chemotherapy after diagnosis of ECD, was noted to be neurologically stable after two years.”
We changed it.
Table on Page 4: The justification is not correct within the cells and several cells appear to have some of the text cut off at the bottom. There also are some discrepancies between what is stated in the table and the text, which I will point out in detail below.
We modified the justification.
Line 160: I take it that this includes pituitary function in the patient with “empty sella”.
Yes, we specified it.
Lines 166-169: Where was the supratentorial enhancement localized in the two patients that had this finding? Either describe it or illustrate it.
We wrote by mistake that there was supratentorial enhancement in both pt 2 and 3, but it was present only in patient 3. We changed the text to correct this. It is specified that the CE in pt 3 was in paratrigonal area.
Lines 170-171: The SWI sequences show iron deposition in areas known to have high iron content. This is not a truly quantitative test, so how do you know it is abnormal?
Expert radiologists rewieved the scans and pointed out the iron accumulation as abnormal for the age of the patients.
Lines 175-176: What is the significance of the “empty sella”? There was DI, which could be related to ECD, but no disfunction of the anterior pituitary is mentioned. Therefore, it is likely that the “empty sella” is not directly related to ECD, i.e., empty sella, without empty sella syndrome.
This comment indeed is correct, and accordingly we changed the text ( currently line 182-183)
Lines 197-198: According to the Table on page 4, the lesions in patient 2 did not have uptake of radioactive tracers. This is important because the tissue diagnosis from the cerebellar biopsy was not called “ECD”.
We reported by mistake in the table column “skeletal involvement” the result of PET scan performed after chemotherapy, which had become normal. The PET scans performed at the time of diagnosis documented abnormal bone uptake of radiotracers. We modified the table accordingly.
Line 199: change “districts” to “sites”
We changed it.
Lines 201-203: This sentence is confusing. If the patient had DI and required desmopressine, how could the hypothalamic-pituitary-axis function be “conserved”?
We specified it (“otherwise conserved”).
Line 207: change “to” to “with”
We changed it.
Line 209: change “at abdomen” to “on abdominal”
We changed it.
Line 210: change “cist” to “cyst”
We changed it.
Line 221: The table on page 4 states that the in Pt 1 the CD207 was negative and does not mention CD117. If CD207 was indeed positive, how often would that occur in ECD and in particular in the face of negative staining for CD1a and S100?
CD117 was negative as well. We changed it in the text.
Line 224: See lines 37 and 86 above
Please see our comment for line 86.
Line 233: change “captation” to “uptake” and change “to” to “with”
We changed it.
Line 235: change “radio” to “radiotherapy” (is there any information on the dose delivered?)
We changed it. Unfortunately, we have not this information from the available medical records.
Line 241: change “history” to “histories”
We changed it.
Line 242: start at new sentence at the colon
We did it.
Line 243: change “at” to “in”
We changed it.
Line 246: change “at” to “in”
We changed it.
Line 251: change “both” to “either”
We changed it.
Line 276-281: Awkward wording. Change sentence to “ This case series illustrates the occurrence of Erdheim-Chester disease presenting exclusively with neurological manifestations.” Then start new sentence with “This issue now appears important, as CNS involvement may affect the life expectancy of ECD patients [19], and recent research indicates there is a significant response of CNS symptoms to novel drugs specifically targeting the MAPK pathway [14,20], the dysregulation of which plays a main role in the pathogenesis of ECD.”
Thanks for your suggestions. We changed it.
Line 286: change “at” to “on”
We changed it.
Lines 291-292: What is the reported age range of onset? Is a cohort of 4 patients adequate to say anything meaningful about a discrepancy?
It was only a descriptive sentence about the age range observed in our small cohort compared to the literature. Since this data is also reported in the results, we can remove it from this section
Line 299: none of your patients had epilepsy, which is a very low prevalence This sentence should be worded better.
We changed it.
Lines 313-315: See Lines 170-171
Please see the comment for lines 170-171.
Line 316: Why is the MRI appearance “peculiar”. I thought it was a subtype of what is expected in ECD (Lines 64-66).
We changed the word in “suggestive”.
Lines 319-320: Which “case” is being referred to? There were two cases with contrast-enhancement described above. (Lines 166-169) Change “remind” to “reinforce” or “support”
Pt 3 (we specified it in the previous comment). We changed “remind”.
Line 322: change “consists in” to “consists of”
We changed it.
Line 325-326: The empty sella is likely to be a red herring here. The presence of DI is the important finding and does not necessarily follow from the empty sella. You would not get an empty sella from damage to the posterior lobe alone.
We agree with you, and we therefore change the sentence in line 334-335.
Line 343: the results for pt 2 need to be reconciled with the Table on page 4 as noted above
We did it.
Lines 355-64: the relationship between “histiocytic sarcoma” and ECD needs to be clarified with some support from the literature. It still is uncertain to me whether Pt 2 has ECD or something different. If that patient is to be included, there needs to be more justification as to why?
We added 3 references, outlined in red in the section
Line 379: I think this should be patient 1 and not patient 2.
In that line ( now line 383) , we referred to pt 3 not 2
Lines 386-391: There need to be citations supporting some of these statements. What is the evidence for neurodegeneration in the absence of infiltration of histiocytes?
We added references and changed the text.
Round 2
Reviewer 2 Report
My criticisms have been adequately addressed. One minor change: in line 300 "According to" should be changed to "In accordance with"